# Correlation between the Perfusion Index and Intraoperative Hypothermia: A Prospective Observational Pilot Study

**DOI:** 10.3390/medicina57040364

**Published:** 2021-04-08

**Authors:** Sangho Lee, Keon-Sik Kim, Sung-Wook Park, Ann-Hee You, Sang-Wook Lee, Yun-Jong Kim, Mihyeon Kim, Ji-Yoo Lee, Jeong-Hyun Choi

**Affiliations:** 1Department of Anesthesiology and Pain Medicine, Asan Medical Center, University of Ulsan College of Medicine, Seoul 05505, Korea; silzzang15@naver.com (S.L.); sweagle@naver.com (S.-W.L.); 2Department of Anesthesiology and Pain Medicine, College of Medicine, Kyung Hee University, Seoul 02447, Korea; keonsikkim@gmail.com (K.-S.K.); demerol@khu.ac.kr (S.-W.P.); 3Department of Anesthesiology and Pain Medicine, Kyung Hee University Hospital, Seoul 02447, Korea; annhee.you@gmail.com (A.-H.Y.); mewusi@hanmail.net (Y.-J.K.); rlaalgus03@gmail.com (M.K.); ekomomai87@daum.net (J.-Y.L.)

**Keywords:** perfusion index, intraoperative hypothermia, general anesthesia

## Abstract

*Background and Objectives:* We examined the association between the baseline perfusion index (PI) and changes in intraoperative body temperature during general anesthesia. The PI reflects the peripheral perfusion state. The PI may be associated with changes in body temperature during general anesthesia because the degree of redistribution of body heat from the central to the peripheral compartment varies depending on the peripheral perfusion state. *Materials and Methods:* Thirty-eight patients who underwent brain surgery were enrolled in this study. The baseline PI and body temperature of the patients were measured on entering the operating room. Body temperature was recorded every 15 min after induction of anesthesia using an esophageal temperature probe. Univariate and multivariate logistic regression analyses were performed to identify the risk factors for intraoperative hypothermia. *Results:* Eighteen patients (47 %) developed hypothermia intraoperatively. The baseline PI was significantly lower among patients in the hypothermia group (1.8 ± 0.7) than among those in the normothermia group (3.0 ± 1.2) (*P* < 0.001). The baseline PI and body temperature were independently associated with intraoperative hypothermia (PI: odds ratio [OR], 0.270; 95% confidence interval [CI], 0.105–0.697; *P* = 0.007, baseline body temperature: OR, 0.061; 95% CI, 0.005–0.743; *P* = 0.028). *Conclusions:* This study showed that low baseline PI was the factor most related to the development of intraoperative hypothermia. Future studies should consider the PI as a predictor of intraoperative hypothermia.

## 1. Introduction

Perioperative hypothermia increases the risk of postoperative morbidity and mortality [1,2,3]. In general, 50–90% of the patients experience perioperative hypothermia [4]. During the first hour of general anesthesia, anesthetic-induced peripheral vasodilation causes redistribution of body heat from the central to the peripheral compartment, leading to a rapid decrease in body temperature [5].

The peripheral perfusion index (PI) is a non-invasive measurement of the perfusion state of peripheral blood vessels. The PI is calculated as the ratio of pulsatile to non-pulsatile signals in arterial blood flow [6,7]. To date, the PI has been used to predict low blood pressure [8], identify early success indicators of central and peripheral nerve blocks [9,10], assess pain [11,12,13], evaluate systemic vascular resistance [14,15], indicate the success of sympathectomy for hyperhidrosis [16], and identify the incidence of hypotension after spinal anesthesia [17]. However, few studies have investigated the correlation between the PI and body temperature.

We hypothesized that the PI may be associated with changes in body temperature during general anesthesia because the degree of redistribution of body heat from the central to the peripheral compartment varies depending on the peripheral perfusion state. Therefore, the purpose of this study was to evaluate the association between the PI and development of intraoperative hypothermia during general anesthesia.

## 2. Materials and Methods

### 2.1. Patients

The Institutional Review Board of Kyung Hee University Hospital (IRB file no. KHUH 2019-03-015, the dates of approval; 10 April 2019) approved this study. After obtaining written informed consent, we enrolled 38 adult patients undergoing elective brain surgery (microvascular decompression, cerebral aneurysm neck clipping, and tumor removal) under general anesthesia between April and July 2019 (Figure 1). We strictly limited the inclusion and exclusion criteria to control for confounding factors that could affect hypothermia during surgery to evaluate the effect of the PI alone. The exclusion criteria were body mass index (BMI) < 20 or >30, age < 19 years or >70 years, American Society of Anesthesiologists physical status (ASA-PS) score ≥ 3, and patients with peripheral vascular disease or rheumatic disease. If the BMI is high or low, depending on the amount of body fat, insulation may be affected, so the range of BMI was set to minimize this. In the case of peripheral vascular disease, it was excluded from the study because there may be disorders in vascular relaxation and contraction caused by the autonomic nervous system. In addition, patients who underwent blood transfusion during surgery were excluded from the analysis. In this study cold blood transfusion may affect body temperature, so even those who participated in the study were excluded from the analysis.

### 2.2. Intervention

Patient’s demographic data were recorded such as age, sex, height, weight, ASA class and past medical history (e.g., diabetes mellitus, hypertension). After entering the operating room, the PI was monitored by attaching a sensor to the patient’s finger (Masimo Radical-7; Masimo Corp., Irvine, CA, USA). The baseline core body temperature was measured using an infrared tympanic membrane thermometer (ThermoScan^®^ 5 IRT6030; BRAUN, Kronberg, Germany). A blanket (full body warming blanket, model no. 30000; 3M, St. Paul, MN, USA) and forced air warming (Bair Hugger, model no. 505; Arizant Healthcare Inc., Eden Prairie, MN, USA) were used to maintain the patient’s body at a set temperature of 38 °C. Approximately 1 min elapsed before the PI graph stabilization. Subsequently, the baseline PI was recorded and assessed every 15 min until completion of the surgery. Electrocardiography, radial arterial blood pressure, oxygen saturation, bispectral index (BIS) and ambient temperature were also measured and recorded. The ambient temperature was measured at a distance of approximately 2 m from the patient to exclude the influence of forced air warming. The level of consciousness was monitored using BIS monitoring sensors (BIS Vista, Aspect Medical System, Minneapolis, MN, USA). If the surgical draping site and the BIS attachment site overlapped, after discussion with the surgeon, it was attached in the correct position as much as possible within the range that did not interfere with the operation. Anesthesia was induced and maintained with propofol and remifentanil. Effect-site concentrations of 2.0–2.5 μg/mL for propofol and 5.0–7.0 ng/mL for remifentanil were set for induction and maintenance of anesthesia. A neuromuscular blockade was induced with rocuronium (0.8 mg/kg). After induction of anesthesia, an esophageal stethoscope with a temperature probe (esophageal stethoscope with temperature sensor; Sewoon Medical Co., Cheonan, Korea) was placed 35–37 cm from the upper incisors, and the core body temperature was monitored every 15 min. After tracheal intubation, an inhaled air heater (Ace Heated Controller; ACEMEDICAL, Goyang, Korea) was connected to the breathing circuit. Mechanical ventilation was provided with an inspired oxygen concentration of 50%, fresh gas flow of 3 L/min, tidal volume of 6–8 mL/kg, end-tidal carbon dioxide concentration of 35–40 mmHg, and a respiratory rate adjusted to 10–15 breaths/min. The criterion for the diagnosis of intraoperative hypothermia was a body temperature of <36.0 °C during the surgery [18,19].

### 2.3. Statistical Analysis

Continuous variables are expressed as mean ± standard deviation (SD), and categorical variables are expressed as number (proportion). Demographic and clinical data of the two groups were analyzed using the Student’s *t*-test, chi-square test, or Fisher’s exact test. Univariate logistic regression analysis was used to explore the factors affecting intraoperative hypothermia and variables with *p*-value less than 0.2 and previously described clinically important factors were included in the multivariate logistic regression analysis. Statistical significance was defined as *p* < 0.05. The IBM SPSS statistical software package for Windows (version 22.0; IBM, Armonk, NY, USA) was used for statistical analyses.

## 3. Results

Written informed consent was obtained from 40 patients, and the analysis was performed on 38, after excluding two patients who underwent blood transfusion during the surgery. The patients were divided into two groups: the hypothermia group who had a core body temperature of <36.0 °C during the surgery and the normothermia group who maintained a core body temperature of ≥36.0 °C. Demographic and intraoperative data of the patients are summarized in Table 1. Among the 38 patients, 18 experienced hypothermia during the surgery. Demographic data, such as age, sex, height, weight, BMI, ASA class, history of diabetes and hypertension, duration of anesthesia, amount of administered fluid, dose of anesthetic agents, estimated blood loss, and ambient temperature during the surgery were not significantly different between the two groups. Baseline PI values (3.0 ± 1.2 vs. 1.8 ± 0.7, *p* < 0.001) (Figure 2) and baseline body temperature (37.1 ± 0.4 °C vs. 36.7 ± 0.4 °C, *p* = 0.010) were significantly lower among patients in the hypothermia group compared to those in the normothermia group and this trend persisted until the end of the surgery (Figure 3). Following univariate logistic regression analysis (Table 2), variables with *p*-values < 0.2 and previously described clinically important factors were analyzed by multivariate logistic regression. Ultimately, the baseline PI and body temperature values were independently associated with the occurrence of hypothermia during surgery (Table 3).

## 4. Discussion

This prospective observational pilot study showed that a low baseline PI was associated with the development of intraoperative hypothermia. Our findings show that PI may be related to body temperature as well as perfusion state implies that the perfusion state may need to be considered in body temperature management. We can consider several reasons why low PI can lead to hypothermia. Early hypothermia during general anesthesia is mainly caused by the redistribution of body heat from the central to peripheral compartments, due to vasodilation following use of the anesthetics [5]. The degree of redistribution of body heat may be affected by the peripheral perfusion state, which differs across patients and results in a difference in gradient between temperature of the central and peripheral compartments [20,21]. Low peripheral perfusion state can lead to low peripheral body temperature, thus lowering the overall core body temperature. Therefore, patients with low PI may have a greater degree of decrease in core temperature due to the large redistribution gradient due to the low peripheral compartment temperature [22]. Second, in this study, the baseline body temperature was already lower in patients with a low baseline PI. Low baseline body temperature per se might produce a low baseline PI in the hypothermia patients. However, our results showed that the association of PI was statistically higher than that of baseline body temperature (Odds ratio; baseline body temperature vs. PI, 0.014 vs. 0.247). Patients with low PI may have a low initial core temperature value due to the already low body temperature. However, considering our findings that PI was the most correlated with hypothermia, PI can be used as a representative value for various factors including peripheral perfusion. It can also be considered that patients with low peripheral perfusion may be less efficient in heat transfer using active air warming.

Abdelnasser et al. [10] reported that the PI is a useful tool to evaluate the success of supraclavicular nerve block. In addition, Eric et al. [16] demonstrated that the PI is an intraoperative marker of successful thoracic sympathectomy. These studies to date with PI suggest that the PI can be considered a non-invasive parameter to assess the state of peripheral blood flow and the autonomic nervous system. However, few studies have reported on the correlation between the PI and body temperature. Kuroki et al. [23] reported that the PI correlated with peripheral temperature and the peripheral-core temperature gradient. These results are consistent with our findings that hypothermia may occur more frequently during surgery due to low PI values.

Previous studies have reported that the risk factors for intraoperative hypothermia are age, BMI, preoperative systolic blood pressure, heart rate, baseline core temperature, ASA-PS score, the type of anesthesia and surgery, the duration of preparation and surgery, and ambient temperature [20,24,25,26,27,28,29]. However, previous studies did not include the PI as a risk factor for intraoperative hypothermia. Kasai et al. proposed a scoring model to predict intraoperative hypothermia [25]. However, they did not consider PI as a predictor. If the PI, which showed the strongest correlation in our study, is included in the predictor, a more accurate predictive score can be generated. In addition, aggressive warming strategies such as pre-warming may be necessary for patients with low PI.

However, our study has several limitations. First, the main limitation is the small sample size. Because our study was a pilot study to find out whether PI is related to body temperature, we sought to control the confounding factor by recruiting patients through strict exclusion criteria. As a result, analysis was performed with a small number of samples. In the future, studies on the validity and clinical applicability of PI with larger numbers of samples will be needed. Second, due to our strict control, the type of surgery and patients were homogenous. The surgery type and patients’ group were controlled to evaluate only the effect of the PI. Future large-scale retrospective or prospective studies on heterogeneous patient groups, including children and elderly individuals, and various surgeries will yield clearer results.

## 5. Conclusions

This prospective observational pilot study showed that a low baseline PI was independently associated with the development of intraoperative hypothermia. Therefore, future studies should consider the PI as a predictor of intraoperative hypothermia.

## Figures and Tables

**Figure 1 medicina-57-00364-f001:**
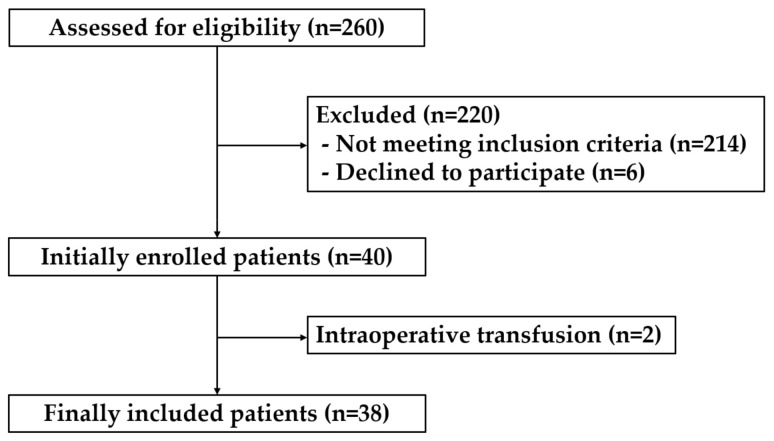
Flow chart of patient enrollment.

**Figure 2 medicina-57-00364-f002:**
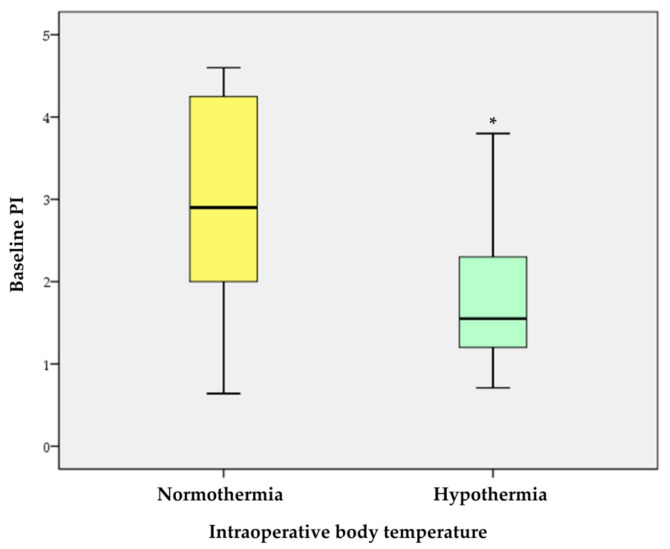
Baseline PI in normothermia and hypothermia groups. * indicates *p* < 0.001.

**Figure 3 medicina-57-00364-f003:**
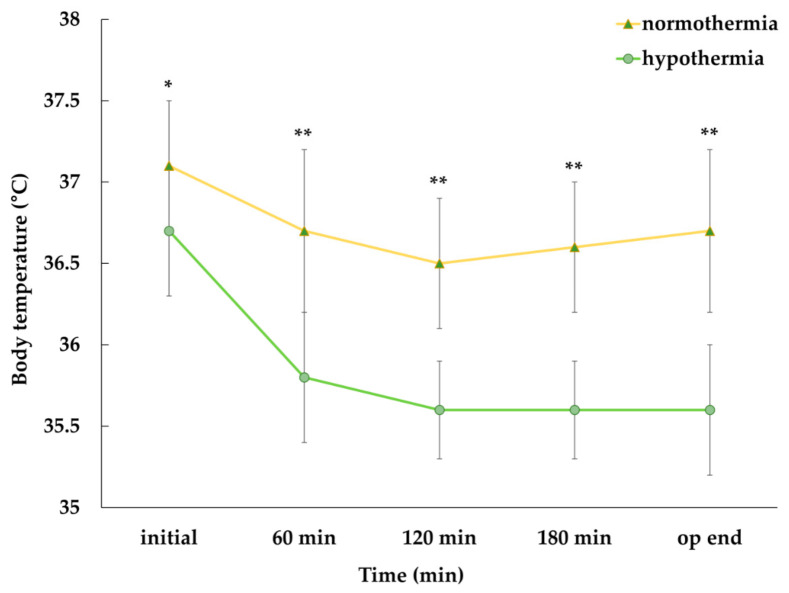
Comparison of core body temperatures between the normothermia group and the hypothermia group. Core body temperatures were significantly higher in the normothermia group at every moment. Data are expressed as mean ± standard deviation. Significant differences between the two groups at each point are indicated as * *p* < 0.05 or ** *p* < 0.01.

**Table 1 medicina-57-00364-t001:** Demographic and intraoperative data of the patients.

	All Patients (*n* = 38)	Normothermia (*n* = 20)	Hypothermia (*n* = 18)	*p-*Value
Age (years)	57.8 ± 7.2	58.5 ± 5.6	57.1 ± 8.7	0.556
Sex (*n* (%))				0.914
Female	25 (65.8%)	13 (65.0%)	12 (66.7%)	
Male	13 (34.2%)	7 (35.0%)	6 (33.3%)	
Height (cm)	160.2 ± 7.9	158.7 ± 7.9	161.8 ± 7.9	0.221
Weight (kg)	62.7 ± 10.7	61.3 ± 11.4	64.3 ± 9.9	0.399
BMI (kg/m^2^)	24.3 ± 2.6	24.2 ± 2.8	24.5 ± 2.4	0.759
ASA (*n* (%))				0.606
I	4 (10.5%)	3 (15.0%)	1 (5.6%)	
II	34 (89.5%)	17 (85.0%)	17 (94.4%)	
Diabetes Mellitus (*n* (%))	8 (21.1%)	4 (20.0%)	4 (22.2%)	1.000
Hypertension (*n* (%))	17 (44.7%)	7 (35.0%)	10 (55.6%)	0.203
Baseline PI	2.4 ± 1.2	3.0 ± 1.2	1.8 ± 0.7	<0.001 *
Duration of surgery (min)	192.1 ± 62.0	194.0 ± 54.7	190 ± 70.8	0.846
Duration of anesthesia (min)	261.8 ± 70.6	260.3 ± 63.5	263.6 ± 79.6	0.886
Type of surgery (*n* (%))				0.370
Aneurysm neck clip	18 (47.4%)	9 (45.0%)	9 (50.0%)	
Microvascular decompression	16 (42.1%)	10 (50.0%)	6 (33.3%)	
Mass removal	4 (10.5%)	1 (5.0%)	3 (16.7%)	
Propofol (mg)	1343.5 ± 366.1	1319.0 ± 308.7	1370.8 ± 428.5	0.669
Remifentanil (mcg)	2888.4 ± 1075.5	3005.9 ± 1052.9	2757.8 ± 1115.5	0.485
Total fluid (mL)	1256.6 ± 444.2	1257.5 ± 411.4	1255.6 ± 490.2	0.989
Estimated blood loss (mL)	305.3 ± 165.9	285.0 ± 175.5	327.8 ± 156.5	0.435
Baseline temperature (°C) ^†^	36.9 ± 0.5	37.1 ± 0.4	36.7 ± 0.4	0.010 *
OR temperature (°C)	20.8 ± 0.8	21.0 ± 0.8	20.6 ± 0.8	0.186

All data are presented as mean ± standard deviation (SD) or number (percentage) of patients. BMI, body mass index; ASA, American Society of Anesthesiologists physical status score; PI, perfusion index; OR, operating room; Normothermia, patients with intraoperative core body temperature ≥ 36.0 °C; Hypothermia: patients with intraoperative core body temperature < 36.0 °C; ^†^ Baseline temperature, tympanic membrane temperature immediately after entering the operating room; * *p* < 0.05 indicating statistical significance.

**Table 2 medicina-57-00364-t002:** Univariate logistic regression analysis of intraoperative hypothermia.

	β	Wals	Odds Ratio	95% Confidence Interval		*p-*Value
Age (years)	−0.028	0.366	0.972	0.888	1.065	0.545
Male	0.074	0.012	1.077	0.281	4.127	0.914
BMI (kg/m^2^)	0.040	0.100	1.041	0.813	1.332	0.752
ASA						
II ^†^	1.099	0.832	3.000	0.283	31.802	0.362
Diabetes Mellitus	−0.134	0.028	0.875	0.184	4.166	0.867
Hypertension	−0.842	1.595	0.431	0.117	1.592	0.207
Baseline PI	−1.270	7.945	0.281	0.116	0.679	0.005 *
Duration of anesthesia (min)	0.001	0.022	1.001	0.992	1.010	0.882
Type of surgery						
Aneurysm neck clip	/	1.751	/	/	/	0.417
Microvascular decompression	−1.099	0.776	0.333	0.029	3.842	0.378
Mass removal	−1.609	1.619	0.200	0.017	2.386	0.203
Baseline temperature (°C) ^††^	−2.140	5.507	0.118	0.020	0.703	0.019 *
OR temperature (°C)	−0.567	1.748	0.567	0.245	1.315	0.186

BMI, body mass index; ASA, American Society of Anesthesiologists physical status score; PI, perfusion index; OR, operating room; ^†^ Odds ratio compared with ASA I; ^††^ Baseline temperature, tympanic membrane temperature immediately after entering the operating room; * *p* < 0.05 indicating statistical significance.

**Table 3 medicina-57-00364-t003:** Multivariate logistic regression analysis of intraoperative hypothermia.

	β	Wals	Odds Ratio	95% Confidence Interval		*p-*Value
Baseline PI	−1.398	6.881	0.247	0.087	0.702	0.009 *
Male	1.839	1.749	6.288	0.412	95.949	0.186
BMI (kg/m^2^)	−0.024	0.013	0.976	0.643	1.482	0.909
Duration of anesthesia (min)	−0.002	0.069	0.998	0.985	1.012	0.794
OR temperature (°C)	−0.559	0.650	0.572	0.147	2.225	0.420
Baseline temperature (°C) ^†^	−4.278	5.198	0.014	0.000	0.549	0.023 *

PI, perfusion index; OR, operating room; ^†^ Baseline temperature, tympanic membrane temperature immediately after entering the operating room; * *p* < 0.05 indicating statistical significance.

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
