# Peer review of "Correlation between the Perfusion Index and Intraoperative Hypothermia: A Prospective Observational Pilot Study"

_medicina, 2021, doi:10.3390/medicina57040364_

Round 1
Reviewer 1 Report
It is important to show more details about the reasons for using those exclusion and iclusion criteria.
Author Response
We would like to thank the editor and reviewers for their thoughtful reviews and appreciate the time and efforts on our manuscript. Based on editor and reviewers’ advice and suggestion, we have revised the manuscript and are re-submitting this manuscript to be considered for publication in Medicina.
Responses to Reviewer #1
1. It is important to show more details about the reasons for using those exclusion and inclusion criteria.
Response: Thanks for the good comment. We have added a description of the exclusion and inclusion criteria according to your comments, in the materials and methods section on page 2, lines 60-71. Inclusion and exclusion criteria were strictly established to minimize confounding factors. If the BMI is high or low, depending on the amount of body fat, insulation may be affected, so the range of BMI was set to minimize this. In the case of peripheral vascular disease that may affect peripheral perfusion, it was excluded from the study because there may be disorders in vascular relaxation and contraction caused by the autonomic nervous system. In addition, patients who under-went blood transfusion during surgery were excluded from the analysis. In this study cold or warmed blood transfusion may affect body temperature, so even those who participated in the study were excluded from the analysis.
Additionally, we increased the resolution of figure 1 and reattached it. Figures have been added to represent the key findings of our study more clearly (figure 2). The figure of body temperature change of the normothermia and hypothermia groups has also been added (figure 3). In the multivariate logistic regression analysis, previously described clinical factors were further added and analyzed and the results were still found to be significant. With your comments, our research paper has been further improved. Thank you.
Reviewer 2 Report
Congrats for your excellent work. It could be improve if you correct Figure 1 (it is blury and I can´t read the text and leyend). And in methods, could you explain how did you monitored level of conciusness.
Author Response
We would like to thank the editor and reviewers for their thoughtful reviews and appreciate the time and efforts on our manuscript. Based on editor and reviewers’ advice and suggestion, we have revised the manuscript and are re-submitting this manuscript to be considered for publication in Medicina.
Responses to Reviewer #2
1. Congrats for your excellent work. It could be improve if you correct Figure 1 (it is blurry, and I can´t read the text and legend).
Response: It seemed that the figure was blurred in the process of inserting it into MS Word. The resolution of the picture has been improved, and the original figure file of dpi 600 is also attached with the extension TIFF. Thank you for your kind comment.
2. And in methods, could you explain how did you monitored level of consciousness.
Response: We added explanation for monitoring level of consciousness, in the materials and methods section on page 3, lines 87-90. The level of consciousness was monitored using BIS monitoring sensors (BIS Vista, Aspect Medical System, USA). If the surgical draping site and the BIS attachment site overlapped, after discussion with the surgeon, it was attached in the correct position as much as possible within the range that did not interfere with the operation. Thank you for the pointed out
Additionally, figures have been added to represent the key findings of our study more clearly (figure 2). The figure of body temperature change of the normothermia and hypothermia groups has also been added (figure 3). In the multivariate logistic regression analysis, previously described clinical factors were further added and analyzed and the results were still found to be significant. With your comments, our research paper has been further improved. Thank you.
Reviewer 3 Report
In this manuscript, the authors describe the association of peripheral perfusion index with the incidence of intraoperative hypothermia. The authors found that the low baseline perfusion index predicts the incidence of hypothermia. In general, the theme in the manuscript is interesting and important for perioperative medicine. However, there are a few major concerns with this manuscript that need to be addressed by the authors.
Major points
- The authors used multivariate logistic regression analysis after several univariate analyses. However, in modern statistics, this strategy is not appropriate for multivariate analysis. The selection of covariates for inclusion in a multivariate model only when they were “statistically significant” in preceding univariate analysis is not preferred. Many journals do not recommend such a method for selecting explanatory variables. Please carefully read the instructions for authors of Annals of Internal Medicine (https://www.acpjournals.org/journal/aim/authors/statistical-guidance). As noted in the instructions, the authors can refer to Sun’s excellent paper: J Clin Epidemiol 1996;49:907. The authors should choose the previously described clinical factors for multivariate analysis even if they do not have a significant difference in univariate analysis. Try to perform logistic regression analysis again.
- The authors should rewrite the Discussion section. First, state key results with reference to study objectives. Second, state interpretation of results from the other relevant evidence. The authors should discuss your data in the "Discussion" section! In the current study, the authors should discuss the difference in the baseline perfusion index between the two groups (Table 1). The reviewer recognized that this difference was very small. Third, state the generalisability of the results. Forth, state limitations of the study.
- The main concern in this manuscript is the difference in the baseline temperature (Table 1). The reviewer is afraid that low baseline body temperature per se produced a low baseline perfusion index in the hypothermia patients. The authors should discuss this issue in the Discussion section.
Also, the reviewer recommends changing several minor points as described below.
Minor points
- Methods. The authors did not state the issue regarding data collection (e.g., age, BMI) in the Methods section.
- Figure 1. The authors should show the higher resolution file.
- Table 1. The data of the baseline perfusion index are the key finding in the study. The authors should show it as a figure.
- The authors may show the temperature trajectory in each group as a figure.
Author Response
We would like to thank the editor and reviewers for their thoughtful reviews and appreciate the time and efforts on our manuscript. Based on editor and reviewers’ advice and suggestion, we have revised the manuscript and are re-submitting this manuscript to be considered for publication in Medicina.
Responses to Reviewer #3
In this manuscript, the authors describe the association of peripheral perfusion index with the incidence of intraoperative hypothermia. The authors found that the low baseline perfusion index predicts the incidence of hypothermia. In general, the theme in the manuscript is interesting and important for perioperative medicine. However, there are a few major concerns with this manuscript that need to be addressed by the authors.
Response: You pointed out 3 major points and 4 minor points. Thanks for the attentive review. Through your comments, we were able to improve our research. Below are the answers to your comments.
Major points
1. The authors used multivariate logistic regression analysis after several univariate analyses. However, in modern statistics, this strategy is not appropriate for multivariate analysis. The selection of covariates for inclusion in a multivariate model only when they were “statistically significant” in preceding univariate analysis is not preferred. Many journals do not recommend such a method for selecting explanatory variables. Please carefully read the instructions for authors of Annals of Internal Medicine (https://www.acpjournals.org/journal/aim/authors/statistical-guidance). As noted in the instructions, the authors can refer to Sun’s excellent paper: J Clin Epidemiol 1996;49:907. The authors should choose the previously described clinical factors for multivariate analysis even if they do not have a significant difference in univariate analysis. Try to perform logistic regression analysis again.
Response: We believe your comments are important and reasonable. In the multivariate logistic regression analysis, previously described clinical factors such as gender, BMI, and anesthesia time were added and analyzed. The results were still found to be significant and were added to table 3. There was a clearer analysis. Thank you.
2. The authors should rewrite the Discussion section. First, state key results with reference to study objectives. Second, state interpretation of results from the other relevant evidence. The authors should discuss your data in the "Discussion" section! In the current study, the authors should discuss the difference in the baseline perfusion index between the two groups (Table 1). The reviewer recognized that this difference was very small. Third, state the generalisability of the results. Forth, state limitations of the study.
Response: Thank you for your kind suggestion. We have rewritten in the discussion section. As advised, we have added discussions and related references to this study data. We have completed a more structured discussion based on your comments (page 6, lines 162 – page 7, lines 212). The changed parts are marked with red letters.
3. The main concern in this manuscript is the difference in the baseline temperature (Table 1). The reviewer is afraid that low baseline body temperature per se produced a low baseline perfusion index in the hypothermia patients. The authors should discuss this issue in the Discussion section.
Response: As you pointed out in the analysis of the study, the baseline core body temperature was different between the two groups. However, the key finding of our study is that several factors in the patient have an influence and are reflected in the PI value. Statistical analysis also showed that PI was more correlated with intraoperative hypothermia than baseline body temperature. These are the key findings of our research. We have added a description of this to the discussion. We have added a description of this to the discussion section on page 6, lines 170-178.
Also, the reviewer recommends changing several minor points as described below.
Minor points
1. Methods. The authors did not state the issue regarding data collection (e.g., age, BMI) in the Methods section.
Response: Thanks for the good opinion. I mentioned data collection such as age, sex, height, weight, ASA class and past medical history (e.g., diabetes mellitus, hypertension) in the materials and methods section on page 2, lines 74-75.
2. Figure 1. The authors should show the higher resolution file.
Response: It seems that the figure was blurred in the process of inserting it into MS Word. The resolution of the picture has been improved, and the original figure file of dpi 600 is also attached with the extension TIFF. Thank you for your kind comment.
3. Table 1. The data of the baseline perfusion index are the key finding in the study. The authors should show it as a figure.
Response: We really appreciate your insightful comments. In figure 2, baseline PI according to hypothermia was newly created as a box plot. This figure gives us a clearer representation of the key findings of our study.
4. The authors may show the temperature trajectory in each group as a figure.
Response: The graph of temperature change in both groups was added to Figure 3. There were statistically significant differences between the two groups at all time points. It helped readers to understand this study. Thanks for the good comment.
Round 2
Reviewer 3 Report
Well done!